# Rational engineering of a thermostable α-oxoamine synthase biocatalyst expands the substrate scope and synthetic applicability

Ben Ashley[1], Sam Mathew[1], Mariyah Sajjad[1], Yaoyi Zhu[1], Nikita Novikovs[1], Arnaud Baslé[2], Jon Marles-Wright[2] & Dominic J. Campopiano [1] ✉

Carbon-carbon bond formation is one of the key pillars of organic synthesis. Green, selective and efficient biocatalytic methods for such are therefore highly desirable. The α-oxoamine synthases (AOSs) are a class of pyridoxal 5'-phosphate (PLP)-dependent, irreversible, carbon-carbon bond-forming enzymes, which have been limited previously by their narrow substrate specificity and requirement of acyl-CoA thioester substrates. We recently characterized a thermophilic enzyme from *Thermus thermophilus* (*Th*AOS) with a much broader substrate scope and described its use in a chemo-biocatalytic cascade process to generate pyrroles in good yields and timescales. Herein, we report the structure-guided engineering of *Th*AOS to arrive at variants able to use a greatly expanded range of amino acid and simplified N-acetylcysteamine (SNAc) acyl-thioester substrates. The crystal structure of the improved *Th*AOS V79A variant with a bound PLP:ʟ-penicillamine external aldimine ligand, provides insight into the properties of the engineered biocatalyst.

Biocatalysts continue to provide useful alternatives to traditional metal and organic catalysts for the synthesis numerous targets molecules at small and industrial scales[1–6]. Once a natural enzyme has been characterised it can be further improved for bespoke applications by the methods of directed evolution and selection[7,8]. The α-oxoamine synthases (AOSs) are pyridoxal 5'-phosphate (PLP)-dependent enzymes that catalyse the irreversible, decarboxylative, Claisen-like condensation of an amino acid with an acyl-CoA thioester, to generate α-aminoketones (Fig. 1A)[9,10]. The AOS family are core enzymes in the biosynthesis of many important biological primary metabolites[11]; 5-aminolevulinic acid synthase (ALAS) from the heme pathway, 8-amino-7-oxononanoate synthase (AONS) from the biotin pathway and serine palmitoyltransferase (SPT) from sphingolipid biosynthesis[12–14]. Some are also found in the pathways of complex natural products such as KtmB from ketomemicin biosynthesis[15]. Many of the AOS enzymes bring together two primary metabolic building blocks, amino acids and fatty acids, but an understanding of the origin of their substrate specificity is still unclear. For this reason, their mechanism, structure and substrate specificity have been studied in great detail over the last seven decades.

One of the earliest known AOS enzymes to be studied was ALAS, and led to the identification of the substrates glycine and succinyl-CoA.

Mechanistic studies identified the role of the PLP and how the amino acid was activated[16–18]. Since then, detailed kinetic studies by Ferreira and colleagues have led to a consensus mechanism[12,19–21] (Fig. 2). A first x-ray structure of a bacterial ALAS revealed the residues involved in substrate binding and catalysis[22]. Similarly, within the biotin pathway the AONS enzyme catalyses the formation of the aminoketone AON from ʟ-alanine and pimeloyl-thioester, either a CoASH type or an acyl-carrier protein (ACP). The x-ray structure of the PLP-bound form of the *E. coli* AONS was the first structure of an AOS to be determined[23]. Further structural and mechanistic studies have also been carried out, which identified intermediates and conformational changes that occur during the catalytic cycle[9,24]. The first enzyme in the biosynthetic pathway of sphingolipids found in all species is SPT which brings together ʟ-serine and palmitoyl-CoA. Bacterial SPTs are canonical AOS soluble homodimers; whereas, SPTs from higher organisms (human, yeast, plants) are multi-subunit, ER-membrane bound complexes whose activity is controlled by small regulatory subunits[14]. Mutations of the human SPT complex have been strongly linked to various human metabolic diseases. A soluble, cytoplasmic SPT was purified from *Sphingomonas paucimobilis*[25] and the subsequent x-ray structure revealed conserved residues between the bacterial and higher order enzymes, as well

[1]School of Chemistry, University of Edinburgh, Joseph Black Building, David Brewster Road, Edinburgh, UK. [2]Biosciences Institute, Faculty of Medical Sciences, Newcastle University, Newcastle upon Tyne, UK. ✉e-mail: Dominic.Campopiano@ed.ac.uk

**Fig. 1 | AOS-catalysed reaction schemes. A** The reaction catalysed by AOS enzymes. **B** Previous work, using cofactor regeneration in a combined AOS-Knorr pyrrole reaction (KPR), chemo/biocatalytic cascade to generate pyrroles[36]. **C** The work of Narayan et al. generating activity with SNAc thioesters by addition of CoA-mimicking pantetheine[39]. **D** This work using engineered *Th*AOS variants and new amino acid and acetyl-SNAc thioester substrates.

**A) Overall reaction catalysed by α-oxoamine synthase (AOS) enzymes**

**B) A cofactor-recycled chemoenzymatic cascade (previous work)**

90%+ yields in 2 hours
20 α-aminoketones
24 pyrroles

**C) Pantetheine-aided reaction using a simple thioester substrate (state of the art)**

Improved yields

**D) Engineered variants with expanded substrate scope, able to use SNAc acyl-thioesters (this work)**

>70 α-aminoketones

Expanded substrate scope
Good yields with SNAc esters
Improved stability and kinetics

**Fig. 2 | The consensus mechanism of wild type and engineered AOS enzymes.** The enzyme resting state is that of an "internal aldimine" or Schiff base, in which the PLP cofactor is covalently bound to the enzyme via a conserved Lys residue ($P_i$ = phosphate). The first chemical step is reversible transimination of the internal aldimine by an amino-acid to generate a PLP:amino acid "external aldimine". In the wild type AOS a conformational change is caused by binding of the acyl-CoA substrate that rotates the external aldimine into the "Dunathan conformation". This permits the catalytic lysine to deprotonate the amino-acid at Cα and generate a reactive PLP:quinonoid intermediate (observed at 490–510 nm). The nucleophilic quinonoid reacts with the electrophilic acyl-CoA thioester in a Claisen-like condensation to form the C-C bond of a β-ketoacid intermediate and eliminates CoASH irreversibly (detected by the DTNB assay). The β-ketoacid is decarboxylated to generate a PLP:α-aminoketone product external aldimine, which finally reacts with the Lys residue to release the α-aminoketone product and return the AOS to the internal aldimine. In blue the engineered *Th*AOS V79 variants produced in this study have been shown to generate the PLP:quinonoid in the absence of the acyl-CoA *substrate*. This expands the substrate scope to allow binding and reaction between a range of amino acids, acyl-CoAs and acyl-SNAc substrates.

as the residues shared by other members of the AOS family[26]. Subsequent studies captured the PLP:L-Ser external aldimine and also probed the formation a key quinonoid intermediate, which is accelerated by the binding of acyl-CoA (Fig. 2)[27–30]. A final AOS member worth noting is KBL (ketobutyrate acetyl-CoA ligase), which is involved in the coupling of glycine and acetyl-CoA to form glycine aminoketone[16]. The aminoketone product retains the carboxylation from glycine but this unstable β-keto acid is rapidly reduced to L-threonine by the threonine dehydrogenase (TDH) complex. An x-ray structure of the *E. coli* KBL, combined with other studies, suggested the mechanistic origin of how this AOS controls product formation[31,32].

Despite their potential as complexity-generating, C-C bond-forming biocatalysts, members of the AOS family not been exploited for synthesis until recently. It appears they have been passed over for more tractable enzyme classes which are easier to engineer, have broader substrate scopes and have no requirement for expensive acyl-CoA substrates[33,34]. In recent years however, AOS enzymes have attracted greater attention as synthetically-useful biocatalysts since they generate α-aminoketones that can be elaborated by synthetic chemistry and/or other biocatalysts. We recently made use of the α-aminoketone products generated by the recombinant form of the *Th*AOS from the thermophilic microbe *Thermus thermophilus*[35]. The α-aminoketones were combined in a chemo-biocatalytic cascade with various β-keto esters in a Knorr pyrrole reaction (KPR), to generate pyrroles in excellent conversions over short timescales (>90% in <2 hours, Fig. 1B)[36]. Also, the AOS enzyme Alb29 from the biosynthetic pathway of the *Streptomyces albogriseolus* natural product albogrisin was recently shown to be able to use L-glutamate and a small group of acyl-CoAs to form seven aminoketones[37]. Additionally, Narayan and colleagues have also exploited another AOS enzyme, *Microseira wollei* SxtA to prepare α-deuterated amino acids in excellent yield and purity. There they used the ability of AOSs to deprotonate and re-protonate α-amino acids in the presence of $D_2O$[38]. The same group have also made progress in understanding and circumventing the SxtA requirement for acyl-CoA thioester substrates, replacing them with acyl-SNAc derivatives (Fig. 1C)[39]. However, the issues of a relatively strict amino acid substrate specificity and reliance on complex cofactor or cofactor-mimicking molecules remain.

In this work, we show for the first time that an AOS biocatalyst (*Th*AOS) can be rationally engineered to use an expanded range of natural and unnatural amino acid substrates. This was achieved by targeting a residue, valine 79 (V79), that is close to the *Th*AOS PLP binding site, but not thought to be involved directly in the catalytic mechanism. Structural studies of AOSs have shown this residue is part of a ~13 amino acid flexible loop that appears to control amino acid substrate binding. We found that substitution of the V79 side-chain opens up entry to a range of natural and unnatural amino acids. Spectroscopic analysis of *Th*AOS V79 variants also provided insight in the intermediates involved. Additionally, the turnover with simple and inexpensive acyl-CoA-mimicking N-acetylcysteamine (NAC or SNAc) thioesters was greatly enhanced with the use of engineered *Th*AOS variants. Using the most thermostable *Th*AOS variant, we coupled this new reactivity in a bio-catalytic cascade with the KPR, removing the reliance on our previous acyl-CoASH recycling system (Fig. 1B) and greatly simplifying the process (Fig. 1D). Furthermore, we determined the crystal structure of the *Th*AOS V79A variant in complex with a PLP:L-penicillamine (L-Pen) external aldimine with a view to understand the impact of engineering at V79. Our study suggests that subtle changes in the active site of an AOS biocatalyst influence the substrate binding and convert these conformationally-dynamic enzymes into a catalytically active state that promotes reaction with a range of non-natural substrates. Our findings should also encourage both rational engineering and random mutagenesis/selection of other members of the AOS family to deliver a range of synthetically useful biocatalysts.

## Results and Discussion
### Substrate binding to wild type ThAOS
The original paper that characterized *Th*AOS showed that it could accept Gly, L-Ala and L-Ser amino acids and acetyl-CoA, pimeloyl-CoA and

palmitoyl-CoA acyl-thioester substrates[35]. It was also shown that *Th*AOS catalysed both KBL (Gly/acetyl-CoA) and AONS (L-Ala/pimeloyl-CoA) reactions. In our previous work, we determined a broader substrate scope of *Th*AOS using a relatively high-throughput kinetic assay which uses the thiol-reactive DTNB reagent (Ellman's) to monitor CoASH release (Fig. 1A)[36]. The *Th*AOS enzyme was active with amino acids L-Ala, Gly, L-Ser and also L-Aba, and could catalyse condensation with acetyl-, propionyl-, butyl-, hexanoyl- and octanoyl-CoA thioester substrates. Therefore, we concluded that the *Th*AOS biocatalyst has an unusually broad substrate scope for an AOS which suggests it could be further expanded with enzyme engineering.

In this new study we took advantage of the inherent UV-vis properties of the PLP-dependent *Th*AOS. Changes in the PLP UV-vis spectrum give insights into substrate binding, as well as the formation of key intermediates during the catalytic mechanism[40,41]. The most intensively studied members of the AOS family have been SPT, ALAS and AONS with a view to understand their substrate specificity within the biosynthetic pathways in which they operate. These combined studies have led to a consensus catalytic mechanism of the AOS family with various lines of evidence to support the structure and role(s) of key intermediates and active site residues (Fig. 2). The AOS enzymes exhibit characteristic PLP spectroscopic properties, with a clear shift in the UV-vis spectrum upon binding of the amino acid substrate to form the PLP:amino acid external aldimine. The binding of the second, acyl-CoA substrate is thought to cause a conformational change (proposed by Dunathan[42]) that allows deprotonation of the proton at C-α to generate the reactive quninonoid/carbanion intermediate. This species subsequently reacts in the C-C forming, Claisen-like condensation step to generate a β-keto acid intermediate, which is readily decarboxylated to form the amino ketone product. The observation of key intermediates, such as the quinonoid, is dependent on the specific AOS/substrate combination.

We studied the binding of *Th*AOS with its amino acid substrates using UV-vis spectroscopy, according to the methods of Webster et al. and Raman et al. that were used to study both *E. coli* AONS and *S. paucimobilis* SPT[9,30]. Recombinant, purified *Th*AOS displays a smaller, broad absorbance between 330–360 nm and a more intense broad absorbance between 380-480 nm corresponding to the tautomeric forms of the PLP-bound internal aldimines (Figs. S1A and 2). Titration of the core substrate Gly (0–16 mM) with *Th*AOS led to a small decrease in the 330 nm peak and a small increase in the 430 nm peak (Fig. S1B). A similar titration of L-Ser (0–16 mM) led to a small increase in the peak at 330 nm and a decrease in the 430 nm peak. With L-Ala (0-16 mM) there was little change in the 330 nm absorbance, a slight decrease and shift (by ~2–4 nm to 426–428 nm) in the 430 nm peak and the observation of a small, broad peak between 490–505 nm. Taken together this spectroscopic data confirms that these substrates bind to the *Th*AOS to form PLP:amino acid external aldimines and that L-Ala binding leads to a small amount of PLP: L-Ala quinonoid formation.

Having established amino acid binding, we then explored what happens upon addition of the second substrate, acetyl-CoA to the external aldimine forms of *Th*AOS. Little change was observed with Gly and L-Ser, but we observed the clear formation of a peak at 501 nm (with a shoulder at 472 nm) with L-Ala as the substrate, suggesting this substrate combination generates an intense quinonoid species (Fig. S2). The observation of such an intermediate suggests subtle differences in the way each substrate binds to and is stabilized by the enzyme. These results suggested that *Th*AOS could potentially bind a greater range of substrates than previously thought – either non-productively to form a *Th*AOS:PLP-amino acid external aldimine, or productively, to form the product aminoketone in the presence of an acyl-CoA substrate (Fig. 2). Moreover, we hoped that the *Th*AOS could be engineered to further expand its substrate scope.

### Rational engineering of ThAOS for an expanded substrate scope
We previously determined the crystal structure of the *Th*AOS homo-dimer with the PLP bound cofactor as an internal aldimine form (PDBs: 7POA, 7POB and 7POC)[36]. In the absence of a substrate bound structure we modelled the PLP:Gly external aldimine from the crystal structure of

**Fig. 3 | Identification of Val79 as a target for enhancing the substrate scope of *Th*AOS.**
**A** Interaction between Val79 and the "sidechain" proton of a Gly substrate in an overlay of the crystal structure of the PLP-bound form of *Th*AOS (PDB: 7POA) with the PLP:Gly external aldimine form of *R. capsulatus* ALAS (PDB: 2BWP). The V79 sidechain of the opposite monomer is predicted to be in relatively close proximity (3.5 Å) to the C-α of the amino acid. **B** A sequence and structural alignment of various AOS biocatalysts highlighting the residues equivalent to position V79 of *Th*AOS. The KBL enzyme from *E. coli* (PDB: 1FC4) retains the valine residue whereas the AONS enzyme from *E. coli* and SPT enzyme from *S. paucimobilis* (PDB: 2JG2) have a serine side chain. An Alb29 enzyme from *S. albogriseolus* (PDB: 8XHA) has a leucine and the ALAS enzyme from *R. capsulatus* has a threonine at this equivalent position (PDB: 2BWP). The full alignment is in Fig. S23.

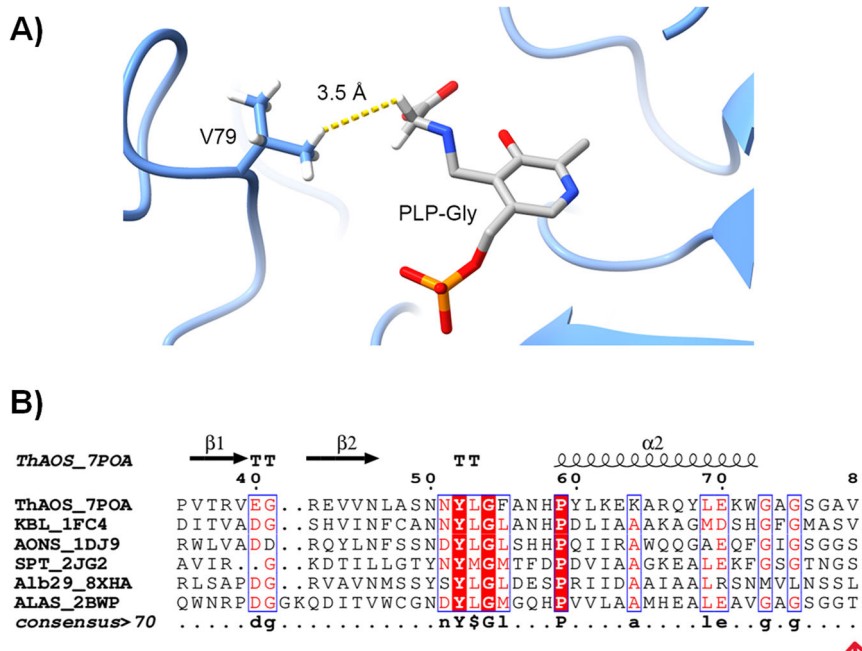

*Rc*ALAS (PDB: 2BWP), a homologue from the AOS family. This model highlighted a close interaction (3.5 Å) between the proposed position of the substrate amino acid sidechain and that of Val79 from the other *Th*AOS monomer within the dimer structure (Fig. 3A). Thus, Val79 appeared to be an important site for amino acid substrate selectivity, and it was targeted for site-directed mutagenesis with the aim of expanding the amino acid substrate scope. This residue is conserved as a Val side chain in all KBL enzymes but replaced by a Ser, Leu and Thr in other AOS enzymes such as SPT, ALAS, AONS and the recently discovered Alb29 (Fig. 3B). This lack of conservation also suggests that changes at this residue might alter the substrate scope, but would not adversely impact the catalytic activity. Our initial hypothesis was that *Th*AOS V79 variants would display an expanded amino acid substrate scope.

**Activity screening of ThAOS V79 variants identifies improved biocatalysts**

Initially, we aimed to reduce size of the Val79 side-chain, ideally permitting the entry of bulkier amino acids besides those already turned over by WT *Th*AOS. As proof of concept, we prepared the enzymes *Th*AOS V79A and V79G as initial variants to investigate their substrate scope (Table S1). The mutant biocatalysts were expressed and purified in the same manner as the wild type *Th*AOS (Fig. S3) and were screened for activity against a panel of amino acids (16 mM) and acetyl-CoA (1 mM), using the colorimetric DTNB thiol detection assay at 50 °C[30,36]. In our screen we included all 20 proteinogenic amino acids, plus a range of unnatural amino acids (UAAs) (Fig. S4).

Of the 33 amino acid substrates tested with acetyl-CoA as the acyl-thioester substrate, *Th*AOS V79A and *Th*AOS V79G displayed activity with 16 amino acids (Table 1). These split into 9 UAAs (L-Aba, DL-Alg, L-Cpa, L-Cpg, L-Hsr, α-Ile, L-Nva, L-Pra and L-Oas) and 7 proteinogenic amino acids (L-Ala, L-Asp, Gly, L-Ile, L-Ser, L-Thr and L-Val). Both variants were significantly faster than the wild type biocatalyst for the original set of four amino acid substrates (L-Aba, L-Ala, Gly and L-Ser), and in general *Th*AOS V79G was faster than *Th*AOS V79A. These variants were kinetically characterised in more detail using the DTNB assay, confirming *Th*AOS V79G to be faster than *Th*AOS V79A in most cases (Fig. S5, Table S2). With the *Th*AOS V79A variant, the catalytic efficiency ($k_{cat}/K_M$) for all four original amino acid substrates was improved substantially (by 2.4x – 64.8x fold). Similarly, the *Th*AOS V79G variant showed improved catalytic efficiency with these four reactions by (2.7x – 57.5x fold), relative to WT *Th*AOS with

both variants displaying the highest activity with L-Ser. Having identified Val79 as a critical residue involved in substrate binding, we attempted to obtain other useful and/or interesting *Th*AOS variants by performing saturation mutagenesis at this position (Table S3). To allow high throughput analysis this expanded set of V79 variants were prepared and expressed in a mini culture format, taking advantage of the thermostability of the biocatalysts.

Under the standard culture conditions, no expression was detected for seven of the possible 17 new variants (*Th*AOS V79D, F, H, N, P, Q and W). Two variants (*Th*AOS V79K and V79Y) expressed but were insoluble, but eight (*Th*AOS V79C, E, I, L, M, R, S and T) were obtained in soluble form (Table S3). These soluble variants were purified from the cell free extract *via* a heating step at 80 °C for 30 minutes, prior to centrifugation (Fig. S3). The variants which were not stable to this treatment were purified by the full Ni²⁺ affinity chromatography (IMAC) method. The purified *Th*AOS variants were screened against the broad range of amino acid substrates with acetyl-CoA and the release of CoASH observed using the DTNB assay. It was clear that the *Th*AOS V79S variant displayed increased activity relative to the wild type *Th*AOS with the core set of four amino acids (L-Aba, L-Ala, Gly and L-Ser), and also accepted the UAAs, L-Cpg and L-Pra (Table 1). Of the other variants, *Th*AOS V79M and *Th*AOS V79R turned over only L-Ser to a lesser extent and *Th*AOS V79T was only active with L-Ala. Amongst the other soluble mutants, *Th*AOS V79C and *Th*AOS V79E displayed no detectable activity with the amino acid substrates tested. The *Th*AOS V79I variant showed the same substrate profile as wild type but with a decreased rate, whilst the *Th*AOS V79L turned over L-Cpg as well as the original four substrates, and at a slightly improved rate than the wild-type biocatalyst. The retention of activity observed with the *Th*AOS V79I/L variants is understandable since these are all similar, hydrophobic side-chains.

We sought to understand the differences in the activities of the *Th*AOS V79 variants by simple consideration of the chemistry of each side chain (Fig. S6). It appears that the presence or absence of a substituent (e.g. methyl) at the β-position of the amino acid side chain influences not only the substrate scope but also the catalytic activity. Furthermore, decreasing the size of the side-chain in the *Th*AOS V79A and *Th*AOS V79G variants also increases activity. It is clear that the residue at this position in the *Th*AOS structure is one of the important determinants of amino acid substrate selectivity. It is also notable that the Val79 residue is not conserved across other members of the AOS family that display differences in their amino

**Table 1 | Substrate screen of *Th*AOS wild type and eight V79 variants**

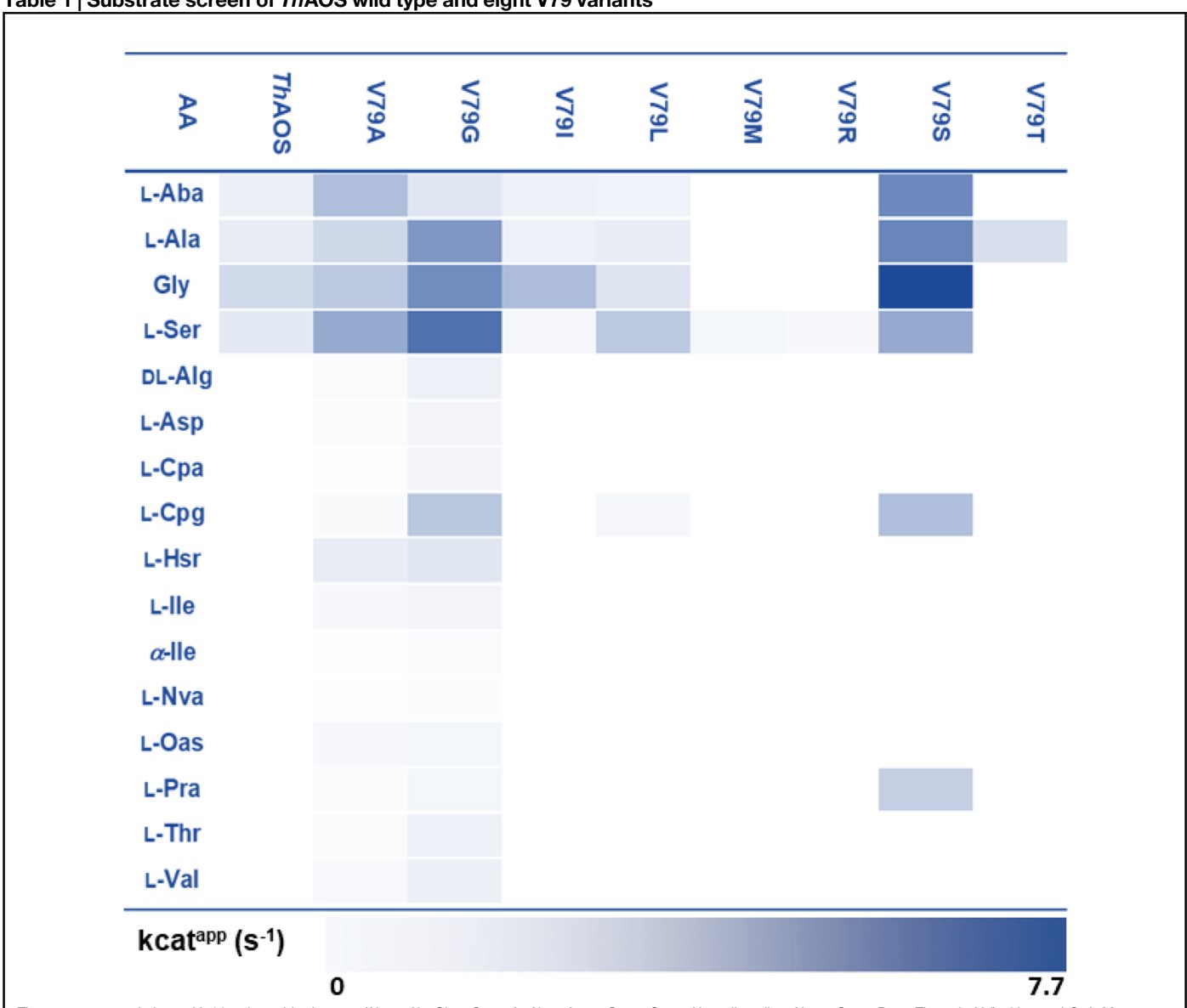

The screen was carried out with 14 amino acid substrates (Aba, ʟ-Ala, Gly, ʟ-Ser, ᴅ-/ ʟ-Alg, ʟ-Asp, ʟ-Cpa, ʟ-Cpg, ʟ-Hsr, ʟ-Ile, α-Ile, ʟ-Nva, ʟ-Oas, ʟ-Pra, ʟ-Thr and ʟ-Val) with acetyl-CoA. Mutants were screened using the DNTB assay at fixed, quasi-saturating substrate concentrations (16 mM amino acid, 1 mM acetyl-CoA). Substrates which were tested but either inactive for all variants or indeterminable include ʟ-Cys, ʟ-Glu, ʟ-Phe, ʟ-His, ʟ-Lys, ʟ-Leu, ʟ-Met, ʟ-Asn, ʟ-Pro, ʟ-Gln, ʟ-Arg, ʟ-Trp, ʟ-Tyr, ʟ-Nle, ʟ-Orn, ʟ-Phg and ʟ-Trl. A scale bar describes the activity, the darkest blue shading in the heat map corresponds to a specific turnover number of 7.70 s⁻¹ (for *Th*AOS V79S with Gly and acetyl-CoA), and white denotes no detectable activity (<0.1 s⁻¹). The ThAOS V79C and V79E displayed no detectable activity and were omitted from the figure.

acid selectivity e.g. in the well characterized *S. paucimobilis* SPT which uses ʟ-Ser as a substrate, this residue is Ser102 (Fig. 3B). The residues in this 13 amino stretch are part of a key flexible loop where the AOS enzymes display conformational flexibility in key residues involved in substrate binding and catalysis. The AOS enzymes have another flexible loop containing the conserved "PATP" motif near the C-terminus that also undergoes conformational change. Our work on the AOS member *S. paucimobilis* SPT captured the key PLP: ʟ-Ser external aldimine complex (PDB: 2W8J) and showed that key arginine residues (R378 and R390) in this loop are mobile. They are also involved in recognition of the carboxylate of ʟ-Ser and help stabilize the key PLP: ʟ-Ser quinonoid intermediate[30].

**Expanded acyl-CoA substrate scope of *Th*AOS V79 variants**. Our next goal was to determine the impact, if any, of variations at Val79 on the acyl-CoA thioester substrate scope. In our previous study, we noted that wild-type *Th*AOS turned over all four of the amino acid substrates (ʟ-Aba, ʟ-Ala, Gly, ʟ-Ser) with acetyl-, propionyl-, butyryl-, hexanoyl-

and octanoyl-CoA[35]. Therefore, we used the DTNB assay to screen the three most promising variants (*Th*AOS V79G, V79A and V79S) derived from the amino acid substrate screen with straight-chain acyl-CoA thioesters of increasing acyl chain length (C2, C3, C4, C6, C8, C10 and C12), as well as benzoyl-CoA (Table 2 and Tables S4-S5).

We were pleased to observe that the acyl-CoA substrate range of *Th*AOS V79G, V79A and V79S was retained across the C2-C8 chain length when compared to wild type *Th*AOS. We noted that the *Th*AOS V79G variant was active with all four core amino acids and C10 and C12 acyl-CoA substrates and could also condense Gly and ʟ-Ala with benzoyl-CoA albeit at very slow rates (Table S4). It was also pleasing to observe that two of the new UAA substrates (ʟ-Cpg, ʟ-Hsr) could also be condensed with acyl-CoA thioesters up to C₁₂ (Table S4). The *Th*AOS V79A variant also featured a slightly improved acyl-CoA substrate scope relative to the WT *Th*AOS (Table S5). The *Th*AOS V79S is distinguishable in that it is the one variant that was able to catalyse condensation of the UAA substrate ʟ-Pra with C2-C8 acyl-CoAs at modest rates. Furthermore, this variant displayed

**Table 2 | Heat-map of the activity of ThAOS V79S with a panel of L-amino-acid/acyl-CoA substrate pairings**

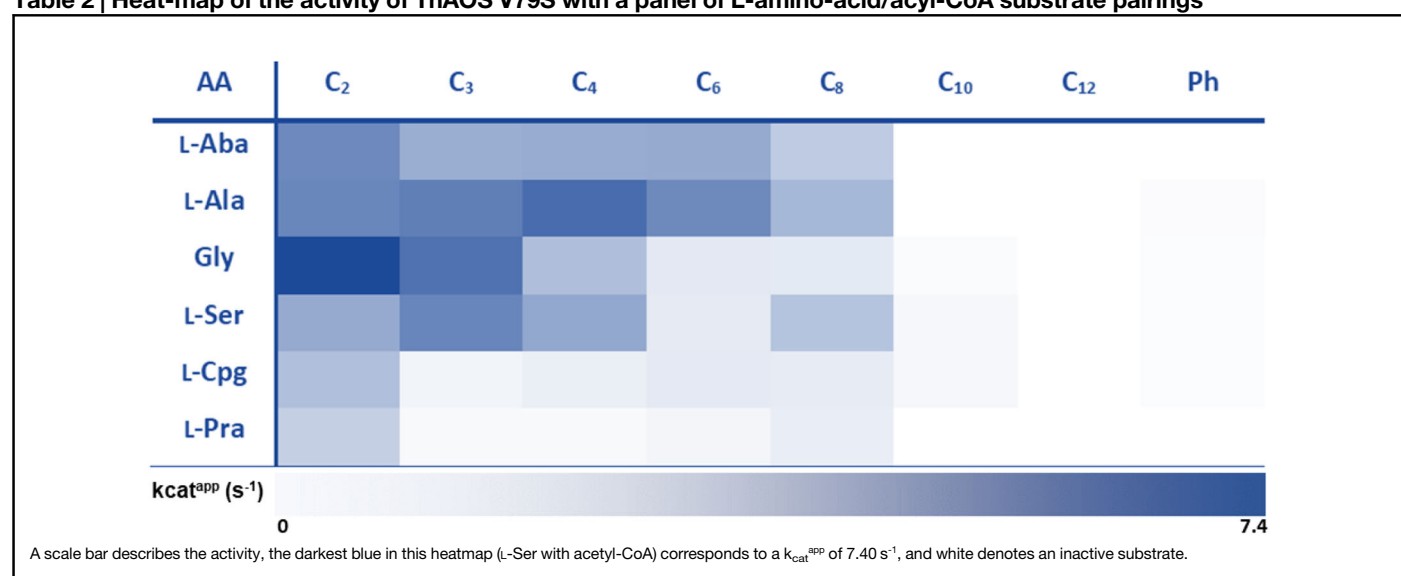

A scale bar describes the activity, the darkest blue in this heatmap (L-Ser with acetyl-CoA) corresponds to a $k_{cat}^{app}$ of 7.40 s$^{-1}$, and white denotes an inactive substrate.

detectable activity with L-Ala, Gly, L-Ser and L-Cpg and benzoyl-CoA (Table 2). Taken together these results show how engineering of *Th*AOS has resulted in active variants with greatly increased substrate scope.

**Acyl-CoA independent quinonoid formation in the *Th*AOS V79G variant.** Since the *Th*AOS V79G variant displayed the broadest activity (Table 1) we have probed its behaviour in the presence of a sub-set of substrates. Here we monitored formation of the key quinonoid intermediate between 490–510 nm to explore the *Th*AOS variants in terms of amino acid substrate binding both alone, and in the presence of the acyl-CoA thioester. The *Th*AOS V79G variant was incubated with Gly, L-Ser and L-Ala and when compared with wild type *Th*AOS we noted similar changes to the UV-vis spectrum when the amino acid was added, although in each case there was a small, but clear absorbance between 480–520 nm indicating the formation of a *Th*AOS V79G PLP:amino acid quinonoid intermediate (Fig. S7 A-C). We also noted that addition of acetyl-CoA to the *Th*AOS V79G PLP:Gly complex led to enhancement of this peak (Fig. S7A). In contrast, addition of the acetyl-CoA to the *Th*AOS V79G PLP: L-Ser complex led to formation of an intense quinonoid signal between 490–520 nm (Fig. S7B), with a shoulder at 468 nm. Moreover, addition of the acyl-CoA thioester substrate to the *Th*AOS V79G PLP: L-Ala external aldimine generated the most intense quinonoid signal between 490–520 nm (Fig. S7C), also with a shoulder 468 nm). Since the *Th*AOS V79G variant also accepted L-Asp, L-Thr and L-Val (Table 1) we looked at quinonoid formation with these three substrates and observed the characteristic formation of the peak around 500 nm of varying intensities for each amino acid (Fig. S7D–F). It is clear that this relatively easy assay is a useful tool to study not only substrate binding, but also formation of this key catalytic quinonoid intermediate.

Quinonoid formation has been studied in only a few members of the AOS family. Studies of the *E. coli* AONS observed quinonoid formation using L-Ala methyl ester (L-Ala-OMe) as a substrate mimic[24]. This ligand bound to the enzyme to form the PLP: L-Ala-OMe external aldimine and upon addition of pimeloyl-CoA two new species were observed by UV-Vis spectroscopy; an intense absorption at 486 nm indicative of formation of the AONS: PLP: L-Ala-OMe quinonoid species and a novel peak at 454 nm which was due to accumulation of the β-ketoacid methyl ester aldimine complex that cannot undergo enzymatic decarboxylation. Taken together, it suggests the clever use of the Me-ester substrate mimic prevents decarboxylation and is a useful probe of the AOS mechanism (Fig. 2). This Me-ester trick was also recently used to stall the SxtA AONS and, in the presence of D$_2$O, allow selective α-deuteration of a panel of 24 natural and UAA α-amino ester substrates[38].

Another elegant study of quinonoid observation is the study of *S. paucimobilis* SPT using an acyl-CoA thioether substrate mimic (S-(2-oxo-heptadecyl)-CoA), a nonreactive thioester analogue of palmitoyl-CoA, to "trick" the SPT PLP: L-Ser complex[29]. This mimic bound to the enzyme and caused a conformational change which led to deprotonation at C-α and PLP: L-Ser quinonoid formation observed at 493 nm. Since the quinonoid could not react with the acyl-CoA thioether the authors used NMR to measure a rate acceleration of C-α deprotonation of more than 100 fold upon binding of the second substrate. This work also supports the proposed general AOS mechanism based on the AOS enzyme *E. coli* AONS[9] (Fig. 2). More in-depth kinetic analyses (both steady-state and stopped flow) of the ALAS enzyme that uses L-Ala and succinyl-CoA have also been reported by Ferreira and colleagues and from these combined studies the key conserved residues involved in AOS catalysis have been proposed[12].

**Thermostability and spectroscopic properties of selected ThAOS variants with expanded substrate scope**
Having obtained a sub-set panel of four competent biocatalyst variants *via* saturation mutagenesis at *Th*AOS V79 (V79A, V79G, V79L, V79S), we set out to further investigate if the mutations had affected the thermostability of the biocatalyst. This was an important consideration in terms of selecting the best *Th*AOS variant that could be used for application in chemical synthesis. Since the *Th*AOS is from a thermophilic bacterium, we decided to study their respective thermostabilities at elevated temperatures (Fig. S8). The wild type *Th*AOS was previously found to be stable to incubation for two hours at 70 °C, so we tested how stable the variants were when incubated at this temperature[36]. The *Th*AOS V70L variant displayed comparable residual activity (92%) to the WT *Th*AOS (89%), while the V79A and V79S variants retained 78% and 79% activity respectively. Unfortunately, the *Th*AOS V79G variant which exhibited the biggest improvement in substrate scope during the screening process, displayed a significant drop in activity (retaining 56.0%) compared to the wild type biocatalyst. We then analysed how stable the variants were upon incubation at 90 °C for 30–120 mins. The WT was to reduced ~15% activity after 120 mins and the V79S, V79G and V79L displayed varying retention of activity (45%, 5% and <5% respectively). However, *Th*AOS V79A retained ~60% activity after incubation at this elevated temperature. By evaluating both the improved substrate scope and thermostability we then decided to take forward the *Th*AOS V79A variant for scaled-up synthesis of a *Th*AOS-derived pyrrole product in combination with the KPR.

**Fig. 4 | Pyrrole production using an acetyl-SNAc substrate and _Th_AOS AOS V79A biocatalyst.** HPLC analysis of the AOS/KPR cascade using _Th_AOS V79A (10 mg mL⁻¹), Gly (32 mM), acetyl-SNAc (32 mM), MAA (32 mM) in HEPES buffer (100 mM, pH 7.5) incubated overnight at 60, 70 and 80 °C. The product pyrrole (**1**) eluting at 14.2 min, the benzoic acid internal standard (**I.S.**) at 13.3 min and acetyl-SNAc substrate at 11.9 min are shown.

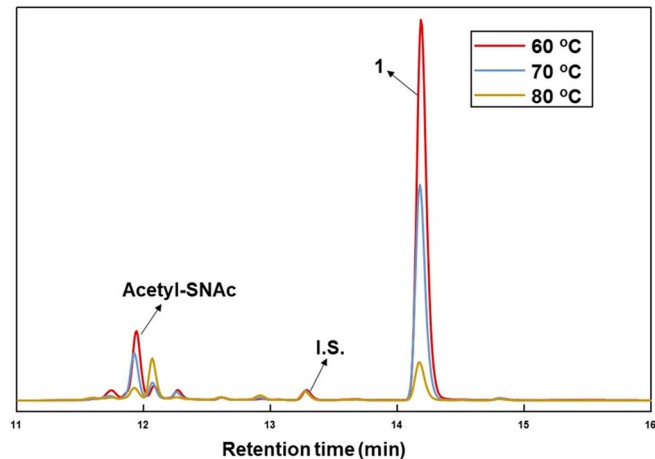

| Yield | 60 °C | 70 °C | 80 °C |
|---|---|---|---|
| _Th_AOS V79A | 35% | 20% | 4% |

**The _Th_AOS V79 variants are viable biocatalysts with alternative, inexpensive acyl N-acetylcysteamine (SNAc) substrates.** Across the AOS family it has been shown that the acyl-CoA thioester substrates display low μM kinetic constants and high turnover rates that are ideal for synthesis. However, the high cost of acyl-CoAs prohibits their use in biocatalysis unless this issue can be overcome by inclusion of an acyl-CoA recycling system. This was achieved in our previous work with wild type _Th_AOS where we used acyl-CoA synthetase (ACS) to convert the CoASH product back into acyl-CoA. Alternatively, a number of simple CoA substrate mimics have been reported that can be used in place of the acyl-CoA which are more affordable and simpler to use[43,44]. Of these, the most widely-used are thioesters of the low-molecular-weight CoA fragment, N-acetylcysteamine (SNAc, Fig. 4). Enzymes capable of using these simple acyl-SNAc thioester substrates in place of an acyl-CoA thioester or pantetheine thioester include thioesterases, ketosynthases, ACP synthases and many others[43,44].

Replacing the expensive acyl-CoA substrates of AOS enzymes with SNAc-thioesters would greatly improve the synthetic potential of these biocatalysts in preparing valuable aminoketone products or being combined into chemo- and bio-catalytic cascades. The only issue that could prevent using these substrate mimics is a kinetic penalty, since the acyl-SNAc substrates normally fail to induce the conformational changes required for quininoid formation in AOS enzymes, tend to display high binding constants and are poor substrates. An AOS homologue, _M. wollei_ SxtA, has recently exhibited improved activity with acyl-SNAcs in the presence of a pantetheine-like auxiliary molecule, which mimics the presence of the acyl-CoA pantetheine arm[39]. Since in many cases the _Th_AOS V79A and V79G variants generate a quinonoid in the presence of the amino acid substrate alone, we hoped that these biocatalysts would react with acyl-SNAc substrates, albeit at high concentrations. Acetyl-SNAc (N,S-diacetylcysteamine) was therefore prepared from cysteamine hydrochloride and acetic anhydride using a published method (see SI for method and Fig. S9A, B for NMR data). The wild type _Th_AOS and three most active variants (V79A, V79G and V79S) were screened for Claisen-condensation activity with acetyl-SNAc against a panel of three amino acids (L-Aba, L-Ala and Gly)

with the DTNB assay at 50 °C (Fig. S10). Wild type _Th_AOS exhibited no detectable activity with any of the substrate combinations. In contrast, we were delighted to observe activity with all of the _Th_AOS variants that were tested. The _Th_AOS V79A displayed activity with acetyl-SNAc and L-Aba and Gly, the _Th_AOS V79G variant was active with L-Aba, L-Ala and Gly and the _Th_AOS V79S variant was also active with all three substrates with Gly displaying a turnover of 0.14 min⁻¹.

The results of this screen allowed us to test the synthetic usefulness of the new reaction using combinations of _Th_AOS variants and amino acids with acetyl-SNAc. To do this, we coupled the AOS biocatalytic reaction with the KPR (at elevated temperature) as we had done previously. The reaction was performed in the presence of methyl acetoacetate (MAA), with the aim of capturing the aminoketone product as a pyrrole, **1** (Fig. 4). The three biocatalysts were screened using high catalyst loadings (10 mg mL⁻¹) with Gly, acetyl-SNAc and MAA (32 mM each) at a variety of elevated temperatures for 16 hours. Reactions were analysed by HPLC as described in our original paper[36]. The results showed that _Th_AOS V79A worked best with Gly and acetyl-SNAc at 60 °C with 35% analytical yield of pyrrole **1** (Fig. 4). We explored if the pyrrole yield could be increased with the higher loading of acetyl-SNAc (32–800 mM). However, no significant increase in yield was observed (Fig. S11).

Since the _Th_AOS V79A variant was the most thermostable under long incubation times at elevated temperatures (Fig. S8) it was used in a preparative scale reaction (10 mL, 32 mM glycine, 32 mM acetyl-SNAc, 32 mM MAA, 15 mg/mL _Th_AOS V79A, 100 mM HEPES, pH 7.5, 16 h). The pyrrole product **1** was extracted and purified and its structure confirmed using NMR and MS (Fig. S12A, B and S13). To confirm that the glycine aminoketone intermediate was formed before conversion to the pyrrole (**1**), it was derivatized using 2,3,4,6-tetra-O-acetyl-β-d-glucopyranosyl isothiocyanate (GITC) and analysed by LC ToF MS (Fig. S14). We also confirmed that other pyrrole derivatives could formed be formed from L-Ala, L-Alg and L-Cpg (Figs. S15–17). Taken together, this result shows that the rationally engineered _Th_AOS V79A variant could be used to prepare various products using inexpensive acyl-thioester starting materials.

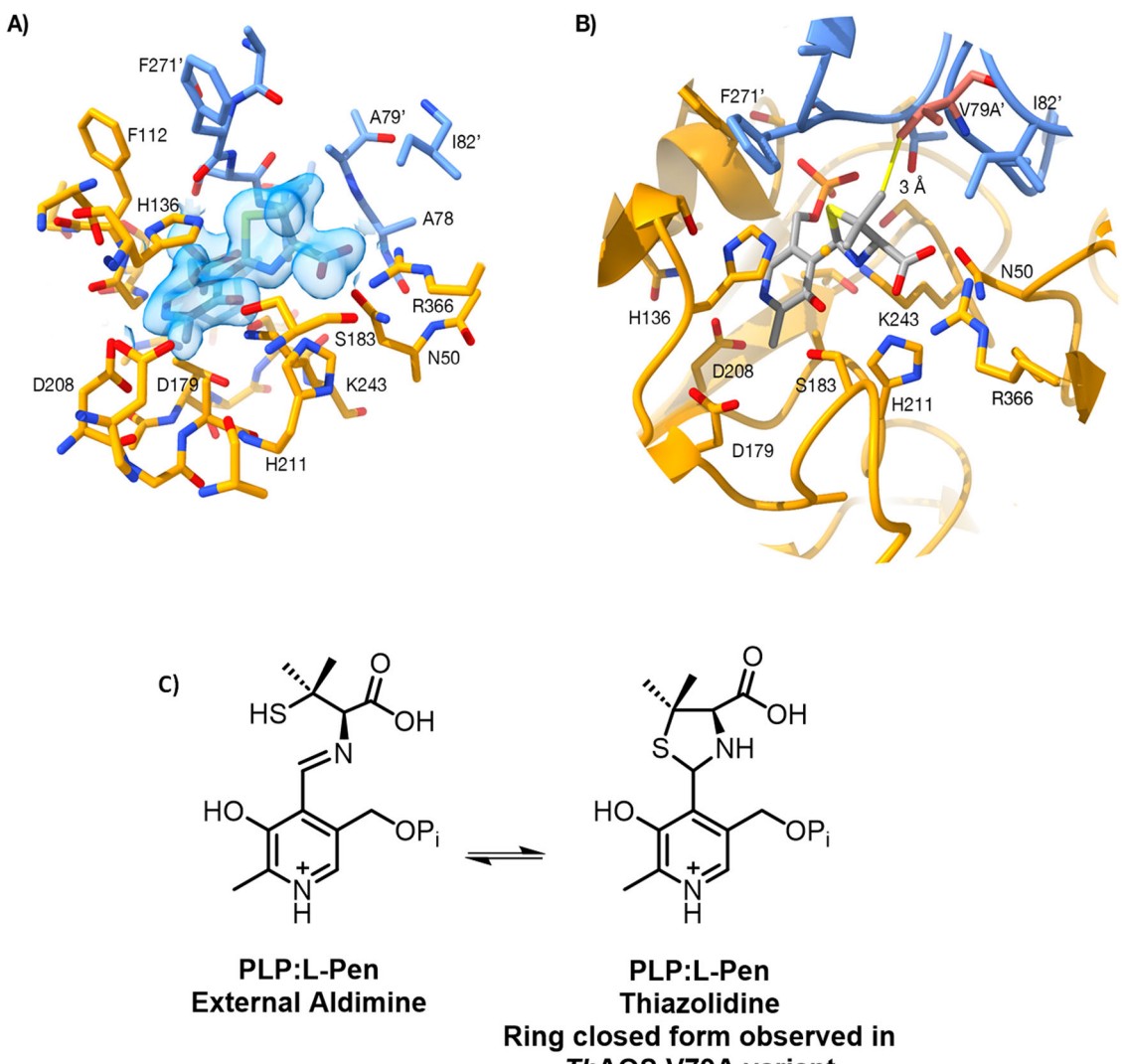

**Fig. 5 | Penicillamine (L-Pen) binding in the *Th*AOS active site. A** Experimental electron density map showing the PLP:L-Pen thiazolidine in the *Th*AOS V79A variant (PDB: 8S1Y). The two protein chains are coloured gold and blue, the side-chain from the other subunit is A79', the ligand is shown with grey carbon atoms and the 2mFo-DFc map is shown as a blue transparent surface at a level of 1σ. **B** Comparison of the *Th*AOS wild-type structure with the V79A variant. The loss of the V79 side chain (shown in pink from the WT structure (PDB: 7POA)), does not cause any gross structural changes at the active site. Removal of V79 opens this region up by around 1 Å, to allow larger hydrophobic substrates to be accommodated. **C** The chemistry of the PLP:L-Pen external aldimine and ring closed PLP:L-Pen thiazolidine observed in the *Th*AOS V79A structure. Figure created using ChimeraX version 1.6.1.

**X-ray structure of the improved, engineered *Th*AOS V79A in complex with the ligand L-Pen.** Inspired by the success of the engineering campaign to deliver improved variants of *Th*AOS at residue V79, we endeavoured to crystallise the variants and solve their crystal structures with bound ligands. This would allow a comparison with the wild type *Th*AOS and might provide molecular insight into the origin of the broadened substrate scope of the engineered variant. Screening revealed that *Th*AOS V79A gave diffracting crystals but we were unable to obtain suitable crystals with various natural amino acid and UAA substrates that came from the screens (Fig. S3). As a final probe to study ligand binding we used the well known PLP enzyme inhibitor L-penicillamine (L-Pen) and monitored the reaction using UV-vis spectroscopy. We used L-Pen since we had studied we had studied this inhibitor in the related the related AOS enzyme *S. paucimobilis* SPT described by Lowther et al. [45,46]. Incubation of L-Pen (1 mM) to the wild type and both *Th*AOS V79A and V79G variants rapidly decoloured both enzymes and led to the formation of a broad peak between 310–380 nm, with an absorbance maximum at 330 nm (Fig. S18A–C). This data is consistent with the formation of a covalent *Th*AOS PLP:L-Pen ring-closed thiazolidine

adduct and this was refined to an excellent fit and geometry. We subsequently used the L-Pen ligand in structural studies and when *Th*AOS V79A crystals were soaked with L-Pen, prior to flash cooling and data collection, clear ligand density was observed in each active site (Fig. 5A). The crystal structure of the *Th*AOS V79A homodimer was determined to 1.5 Å resolution in P1 space group using the wild type *Th*AOS structure (PDB: 7POA) as a molecular replacement model. The data collection and refinement statistics are shown in Table S6.

The structure of *Th*AOS V79A PLP:L-Pen complex (PDB: 8S1Y) shows the L-Pen amine is covalently linked to the PLP cofactor, and the substrate carboxylate is chelated by Arg366 (Figs. 5A and S19A, B). The five-membered ring of the inhibitory thiazolidine intermediate is clearly resolved, and sits in the position of an amino acid substrate sidechain. Most interesting is the distinction between WT *Th*AOS and the engineered, more active variant, *Th*AOS V79A. For this reason, the structure of *Th*AOS internal aldimine was overlaid onto the structure of L-Pen-bound *Th*AOS V79A. The two chains of the *Th*AOS V79A variant align with around 0.4 Å RMSD Cα to the wild type *Th*AOS structure over the 397 residues. There is no significant structural change in the active site in the V79A variant with

PLP: L-Pen bound when compared to the wild type PLP-bound *Th*AOS, with no shift in the loop around the altered residue. The distance between A79 and the thiazolidine methyl group and is 4 Å and modelling the distance to the native V79 gives a distance of 3 Å to the thiazolidone (Fig. 5B). It appears that the removal of the valine side chain increases the volume of the ligand binding site without any large-scale remodelling of this region. Despite a long history of use as an inhibitor of PLP-dependent enzymes, this is the first published crystal structure of a PLP enzyme in complex with L-Pen. However, searching the PDB for the L- and D- forms of the penicillamine ligand identifies two unpublished structures of cysteine desulfurase (CSD) enzymes with structural homology to *Th*AOS with bound penicillamine ligand: NifS from *Helicobacter pylori* (PDB ID: 7XES); and SufS from *Bacillus subtilis* (PDD ID: 7XEN). There is also a structure of the SufS without penicillamine (PDB ID: 7XEN). A detailed discussion of the structural similarity of these complexes is in the supplementary information (Fig. S20–23).

## Conclusion

There is a growing application for biocatalytic routes to both commodity chemicals, as well as high value intermediates and pharmaceuticals[47]. A key step in organic synthesis is C-C formation and a number of enzymes have been developed as biocatalysts. Databases such as RetroBioCat have allowed route developers to plan synthetic strategies using well characterized biocatalysts whose substrate specificity, catalytic rates and methodologies are well curated[48,49]. It is essential to continue to find new natural biocatalysts that can be enhanced by rational engineering and/or directed evolution/ selection. Alternatively, protein scaffolds can be engineered to deliver biocatalysts with no known biological equivalent e.g. the Morita Baylis Hillman reaction[50].

The thermophilic PLP-dependent biocatalyst *Th*AOS catalyses an irreversible, Claisen-like, C-C bond-forming reaction that yields useful aminoketone building blocks[36]. Studying the active site structure of this PLP-dependent biocatalyst, combined with knowledge of the enzyme mechanism, has permitted the generation of variant biocatalysts with significantly improved properties. A single active site residue V79 appears to control access to the active site and conversion to smaller residues such as Gly and Ala allows alternative amino acids and UAAs to bind, without compromising the catalytic activity. Saturation mutagenesis, complementing the original rational mutagenesis strategy, allowed variation at this residue to be fully explored. The best mutant catalyses a total of 71 unique condensations between various amino acid and acyl-thioester substrates, significantly greater than any other AOS previously reported. Furthermore, the ability of the hyperactive mutants to accept simple and affordable SNAc thioesters to a useful degree without the need for auxiliary CoA-mimicking compounds is also a first. We demonstrated the usefulness of this reaction by generating a pyrrole by combining the thermo-stable *Th*AOS variant in a KPR at elevated temperatures. A key determinant in AOS catalysis is the ability of the enzyme to catalyse the formation of a key PLP-bound external aldimine that subsequently generates the key PLP:amino acid quinonoid nucleophile in the present of the acyl-CoA substrate. A convenient UV-vis screen can rapidly identify hit variants from a library. The *Th*AOS V79 variants can generate this reactive species in the absence of the acyl-thioester and can also use truncated cysteamine-derived substrates. In essence we have changed the catalytic mechanism to being acyl-CoA independent. It is clear that this V79 residue, which is found on a dynamic loop, plays an important role and recent work on the related Alb29 AOS suggests other residues in this part of the enzyme could be modified to further expand the substrate scope of AOS[51].

The crystal structure of the *Th*AOS V79A variant with a PLP: L-Pen inhibitor bound displayed little change compared to the wild-type enzyme so this suggest that the enhanced properties must be due to subtle and dynamic changes to the structure and electronics of the PLP-bound transition states. This merits future study on the AOS family of enzymes by detailed kinetics, molecular dynamics and modelling. This work opens the door for the exploitation of members of the expanding AOS family as synthetically useful C-C bond-forming biocatalysts, in the same vein as aldolases, transaminases, racemases, and other widely-used PLP-dependent enzymes[52].

## Methods
### Materials
Commercially available standards, solvents and reagents were purchased from Avanti Lipids, Fluorochem, Sigma Aldrich, Cambridge BioScience and Thermo Fisher Scientific and were used without any further purification.

### Enzyme Expression and Purification
**Expression of *Th*AOS constructs.** A single colony of *E. coli* BL21 (DE3) cells containing a pET28a plasmid encoding *Th*AOS wild type and V79 variants with a TEV-cleavable N-terminal His6 tag was used to inoculate a 5 mL overnight culture of L.B. containing 30 ug/mL kanamycin. After overnight shaking at 37 °C the culture was used to inoculate larger cultures of 1L L.B. media in 2 L Erlenmeyer flasks, which were then grown with shaking at 180 rpm at 37 °C until the OD500 was 0.6-0.8. The culture was then induced with 0.25 mM IPTG overnight at 16 °C. Cells were harvested by centrifugation and cell pellets were stored at -20 °C.

### Heat Purification
Cell pellets were resuspended in HEPES buffer (20 mM HEPES, 150 mM NaCl, 5% glycerol, pH 7.5, ~30 mL) before being sonicated for 15 minutes 30 s on/30 s off. Cell debris was pelleted with ~10,000 x *g* centrifugation for 45 minutes. The clarified cell lysate was next heated in a water bath at 80 °C for 30 minutes with monitoring by SDS-PAGE. After all the *E. coli* proteins had precipitated, precipitate was pelleted once more with another centrifugation step. The solution was then filtered, protein concentration was determined using the Bradford assay and was used without further purification.

### Immobilsed metal affinity purification (IMAC) Ni²⁺ purification of *Th*AOS
Cell pellets were resuspended in HEPES buffer (~30 mL) before being sonicated for 15 minutes 30 s on/30 s off. Cell debris was pelleted with ~10,000xg centrifugation for 45 minutes. The mixture was then loaded onto a 5 mL G.E. Healthcare HisTrap FF column prior to a gradient elution/ fractionation step with Nickel Elution Buffer (HEPES buffer + 500 mM imidazole). The purest fractions (determined by SDS-PAGE) were then buffer-exchanged back into HEPES buffer and enzyme concentration was determined using the Bradford assay.

### Full Purification for structural studies
Cell pellets were treated as according to the Ni²⁺ IMAC Purification, but after the IMAC step yellow fractions were pooled and loaded onto a pre-equilibrated Superdex S200 column, before elution and fractionation with 120 mL HEPES buffer, yielding highly pure mutant enzyme. Samples were concentrated to 20–50 mg/mL stocks, flash-frozen in liquid $N_2$ and stored at -80 °C.

### Enzymatic assay
The activity assay of *Th*AOS and its variants were performed using DTNB assay. The reactions were performed in 60 µL scale using amino acid (16 mM), acetyl-CoA (1 mM), *Th*AOS/*Th*AOS variants (0.1-1 mg mL⁻¹) and DTNB (1 mM) in HEPES buffer (20 mM, pH 7.5) at 50 °C. The UV-vis readings were recorded using a BioTek Synergy HT microplate reader.

### Chemo-biocatalytic *Th*AOS/Knorr Pyrrole Reaction (KPR) Cascades
The *Th*AOS (0-3 mgmL⁻¹) biocatalyst was mixed with amino acid (16-32 mM), methyl acetoacetate (MAA, 32 mM from a 3.2 M stock in MeCN), sodium benzoate (1 mM, internal standard) and acyl-CoA (0–2 mM) or acetyl-SNAc (0-32 mM from a 100× concentrated stock in MeCN) in HEPES buffer pH 7.5 buffer at a final volume of 200 µL in an Eppendorf

tube. Reactions were performed in a Grant-Bio 24-well thermoshaker with shaking at 250 rpm at 30–90 °C for 0–24 h. Reactions were initiated by addition of acyl-CoA. Reactions were terminated by addition of 1 volume of MeCN, and centrifugation for 10 minutes at 13,000 $g$. The supernatant was then analysed by HPLC.

## Scaled up synthesis of pyrrole (1)

A preparative scale reaction (10 mL, 32 mM glycine, 32 mM acetyl-SNAc, 32 mM MAA, 15 mg/mL *Th*AOS V79A, 100 mM HEPES, pH 7.5, 16 h). A 50 uL NAOH (5 M) was added to the reaction mixture, followed by three washes of ethyl acetate. The final product was purified through flash chromatography (50:50 hexane: EtOAc). The pyrrole product 1 was extracted and purified, and its structure was confirmed using NMR (Fig. S12A, B). The 1H and 13 C NMR spectra were recorded on a Bruker Avance III 500 MHz or a Bruker CryoProbe Prodigy 500 MHz, and the solvent was $CDCl_3$.

## UV-Vis spectroscopy

The UV-vis spectrum of the wild-type *Th*AOS and V79 variants in the PLP bound state and in the presence of amino acid and acyl-CoA substrates were recorded on a Cary 50 UV vis spectrophotometer with 1 cm pathlength cuvettes. The spectrophotometer was blanked against buffer, and absorbance intensity was recorded between 250–600 nm on the Fast setting. For titrations, substrate was added to the cuvette from a 100 mM stock in buffer, mixed by pipetting, and allowed to equilibrate for 30 seconds before recording of the new spectrum. The dilution was accounted for in the final spectra by amplifying the new spectra by the dilution factor caused by the addition of the substrate. The whole spectrum was captured and changes in the absorbance maximum of the dominant peaks were plotted and analysed in OriginLab 2019.

## Protein crystallisation

Crystallisation of recombinant *Th*AOS V79 mutants was initially screened using commercial kits (Molecular Dimensions and Hampton Research). Protein concentration was 20–25 mg.mL$^{-1}$. The drops, comprising 0.1 or 0.2 μL of protein solution plus 0.1 μL of reservoir solution, were set up using a Mosquito crystallisation robot (SPT Labtech). The experiments were incubated at 20 °C. Initial hits were of good size, single and could be directly tested. Whilst hits were found in Index (Hampton Research), three conditions were found in Morpheus (A8, A12 and C8, Molecular Dimensions) to lead to alternative crystal space groups. *Th*AOS V79A crystallised in P1 (30 mM sodium nitrate, 30 mM sodium phosphate, 30 mM ammonium sulfate, 100 mM HEPES/MOPS pH 7.5, 12.5% (w/v) PEG1000 and 12.5% (w/v) PEG3350), P21 (30 mM magnesium chloride, 30 mM calcium chloride, 100 mM HEPES/MOPS pH 7.5, 12.5% (w/v) PEG1000 and 12.5% (w/v) PEG3350) and in P212121 (30 mM magnesium chloride, 30 mM calcium chloride, 100 mM Tris/bicine pH 8.5, 12.5% (v/v) MPD, 12.5% (w/v) PEG1000 and 12.5% (w/v) PEG3350). The samples did not require optimisation of additional cryo-protection.

## Synthesis of N,S-diacetylcysteamine (acetyl-SNAc)

Cysteamine hydrochloride (2.11 g, 18.8 mmol) was dissolved in water (20 mL) at 0 °C, and the pH was adjusted to 8.0 with aqueous KOH (8 M). Acetic anhydride (5.72 g, 56.1 mmol) was next added dropwise, and the pH was adjusted to 7.0 with aqueous KOH (8 M). The solution was stirred at 0 °C for 90 minutes, until addition of a drop to a solution of Ellman's reagent DTNB did not cause a colour change. The solution was then extracted with $CH_2Cl_2$ (3×). The organic phase was then washed with acidified water (3×), dried with magnesium sulfate and concentrated by rotary evaporation to afford the title compound as a viscous and colourless liquid (0.156 g). NMR data is shown in Fig. S9A and 9B.

**1H NMR** (500 MHz, $CDCl_3$): δH 5.95 (1H, broad s, NH), 3.46 (2H, t, J = 10 Hz, $-NHCH_2CH_2S-$), 3.04 (2H, t, J = 10 Hz, $-NHCH_2CH_2S-$), 2.37 (3H, s, $-SC(O)CH_3$), 1.99 (3H, s, $-NHC(O)CH_3$).

**13 C NMR** (101 MHz, $CDCl_3$): δC 196.3, 170.3, 39.6, 30.7, 28.9, 23.1.

## Data Collection, Structure Solution, Model Building, Refinement and Validation

Diffraction data were collected at the synchrotron beamline I04 of Diamond light source (Didcot, U.K.) 07/07/2021 at a temperature of 100 K. The data set was integrated with XIA2[53]. using XDS[54] and scaled with Aimless[55]. The space group was confirmed with Pointless[56]. The phase problem was solved by molecular replacement with Phaser[57] using *Th*AOS structure as the search model (PDB: 7POA)[36]. The model was refined with refmac[58]. The PLP: L-Penicillamine ligand was generated using JLigand[59] and optimised with AceDRG[60]. Manual model building with COOT[61] was intercalated between refinement rounds. The models were validated using Coot and Molprobity[62]. Other software used were from CCP4 cloud[63] and CCP4 suite[64]. Figures were made with Chimerax[65]. Data collection processing and refinement statistics are presented in Table S7.

## Reporting summary

Further information on research design is available in the Nature Portfolio Reporting Summary linked to this article.

## Data availability

Protein structure raw data files (MTZ and PDB for 8S1Y) are available from the author. Raw data files for HPLC, NMR, kinetic and UV-vis data are also available upon request.

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

## Acknowledgements
The authors thank the BBSRC for an EastBio (BB/J01446X/1) PhD studentship (B.A.) The BBSRC is thanked for grant funding awarded to D.J.C. (BB/T016841/1) to support S.M. The University of Edinburgh and the Derek Stewart Charitable Trust is thanked for PhD studentship funding (M.S.).

## Author contributions
B.A. and D.J.C. designed the project with respect to engineering the *Th*AOS biocatalyst and isolating improved variants. B.A. designed, expressed and purified the *Th*AOS variants. B.A. also assayed the activities with the reported substrates. Y.Z. prepared and characterized *Th*AOS V79 variants. S.M. carried out characterization of the *Th*AOS V79 variant and scaled up reactions. M.S. and N. N. prepared the acyl-SNAC substrate and carried out the isolation of pyrrole targets. A.B. isolated crystals of the *Th*AOS V79A PLP:L-Pen complex and acquired data from the Diamond Light Source. A.B. and J.M.-W. solved the structure of *Th*AOS V79APLP: L-Pen complex and deposited the structure in the Protein DataBank (PDB). B.A. wrote the initial manuscript. All authors made written contributions to the manuscript and prepared figures and tables. D.J.C. edited and wrote the final version of the paper.

## Competing interests
The authors declare no competing interests.
