## [Transparent Peer Review file · Communications Chemistry]

Rational Engineering of a Thermostable α -Oxoamine Synthase Biocatalyst Expands the Substrate Scope and Synthetic Applicability.

Corresponding Author: Professor Dominic Campopiano

Version 0:

Reviewer comments:

Reviewer #1

(Remarks to the Author)

In this manuscript, the authors demonstrated the structure-guided engineering of ThAOS to achieve variants with broader scope of substrates. The crystal structure of the ThAOS V79A mutant with a bound PLP:penicillamine external aldimine ligand, was investigated. I find it interesting.

I recommend to revise some points (see below).

1. Do the ketone products racemize under the reaction conditions?
2. In the evaluation of the substrate scope, the HPLC and MS analysis of the products with various substrates should be provided. At least some of the products should be purified and provide the NMR data.
3. It would be more clear if you provide a detailed table of content at the beginning of the supporting information.
4. All instances of L or D in front of amino acid should be capitalized and adjust to the smaller size. For example, in line 272. Please check the whole manuscript.
5. Please mention the reference for each method used in this study.
6. Please check the text format in the Method section. The sub-title should be capitalized for each word. The superscript and subscript format in NMR section should be corrected. The NMR data assignment to the specific position should be marked in the structure.
7. The description of materials in the method section is obscure. Please reveal more detailed information.

Reviewer #2

(Remarks to the Author)

The manuscript 'Rational engineering of a thermostable α -oxoamine synthase biocatalyst expands the substrate scope and synthetic applicability' describes the application of PLP-dependent enzyme ThAOS in the biocatalytic decarboxylative Claisen-like condensation reactions between amino acid and thioester substrates. This article builds off of the Campopiano lab's previous studies adapting a CoA cofactor recycling system and chemoenzymatic production of α -aminoketones and pyrroles. A variety of interdisciplinary techniques are used to better understand this C-C bond forming enzymology, including: UV-spectroscopy, substrate engineering, site-saturation mutagenesis, X-ray crystallography, biocatalysis and chemoenzymatic synthesis. I appreciated the effort put into this manuscript and found it to be a very dimensional study into a chemically useful enzyme family. One area of particular interest was the spectroscopic variation of the different mutants on improving the quinonoid species in the presence of the second thioester substrate. I have a few suggestions to improve this manuscript – however, following these adjustments, I believe this manuscript is appropriate for publication in Communications Chemistry.

Major comments:

DTNB assay vs. product output – The authors have nicely shown the use of the DTNB reagent to quantify free sulfhydryl groups and show amino acid and CoA/SNAC substrates that work with ThAOS and mutants. However this does not directly measure α -aminoketone product formation and could result from non-specific hydrolysis of the thioester. Have the authors quantified the DTNB output compared to product formation? Alternatively, has this analysis been done before for this enzyme family in literature to show that the DTNB read out is an appropriate substitute for direct product quantification? Some insight here would help reinforce the utility of this method and give additional confidence to the results for all ThAOS variants presented throughout.

Figure 5 alignment – It took me a while to realize that the different temperature traces were staggered at an angle – I would

encourage the authors to line things up on the same x-axis or use shading to make it obvious which peaks correspond to the acetyl-SNAc, I.S., and 1 in each trace.

Minor comments:

Page 1, line 38 – It felt a bit heavy handed associating refs 15-32 with one sentence in the introduction – potentially breaking them up would be more helpful to delineate their relative contributions.

Figure 1 – Including a citation to sections B and C would further connect them to their literature references. I enjoyed the use of the red highlight for the new C-C bond in Panel A – continuing that in panels B through D would further emphasize the impact of this enzymology.

Page 3, lines 93-96 – Please define KBL, AONS, DTNB, SPT, and other acronyms on their first introduction to help improve transparency to non-expert audiences.

Figure 2 – There is another mechanistic step between the b-ketoacid intermediate and the product aldimine – either depict the enolate intermediate (and a-carbon reprotonation) or show more than one arrow between these intermediates. Quinonoid is also misspelled in the figure legend.

Figure 3, line 185 – the PDBID for ThAOS is missing in the figure caption.

Page 9, Table 1 – Please include a legend for the grey scale shading in the heat map to give insight into the turnover numbers for each variant/amino acid pair.

Page 9, lines 368 and beyond – Only include integer numbers for yields

Page 10, line 388 – Presumably missing reference here in place of (Ashley)

Page 13, lines 490 – 503 – The alpha center stereochemistry for L-penicillamine is not depicted in panel C, and the Figure caption for this panel is missing.

References – Refs 35 and 36 are duplicated

Reviewer #3

(Remarks to the Author)

Please see the attached review file.

Reviewer #4

(Remarks to the Author)

I co-reviewed this manuscript with one of the reviewers who provided the listed reports. This is part of the Communications Chemistry initiative to facilitate training in peer review and to provide appropriate recognition for Early Career Researchers who co-review manuscripts.

Version 1:

Reviewer comments:

Reviewer #2

(Remarks to the Author)

I appreciate the authors' responsiveness to the reviewers comments and believe that the manuscript is stronger and more rigorous as a result. All of my major comments (DTNB assay vs. product output; previous Figure 5 – new Figure 4 alignment) have been appropriately revised and I appreciate the discussion around the first point. However, many of my original minor comments have either been selectively addressed or not corrected at all. I encourage the authors to update the following items prior to further consideration.

Figure 1: I like the inclusion of the red highlighted bond in all A-D sections to show the new connection and appreciate its inclusion. I would encourage the authors to cite the work for sections B and C either on the figure, in the figure legend, or both, to give better connection to their literature references.

Figure 2: My original comments were not addressed – I'm including them again here:

There is another mechanistic step between the b-ketoacid intermediate and the product aldimine – either depict the enolate intermediate (and a-carbon reprotonation) or show more than one arrow between these intermediates. Quinonoid is also

misspelled in the figure legend.

Table 1: The color change from grey to blue is fine, however I don't see a scale based on enzyme activity as suggested by the authors. Including my original comment again:

Please include a legend for the grey (now blue) scale shading in the heat map to give insight into the turnover numbers for each variant/amino acid pair.

A similar scale should be included in Table 2 for improved transparency

Figure 5C: The alpha stereocenter for L-penicillamine is still missing – please correct. The figure caption is present now, thank you for including it.

Reviewer #3

(Remarks to the Author)

The authors have provided a comprehensive and detailed response to my and other reviewers initial feedback, addressing each of my concerns thoughtfully and with clarity. Their revisions have significantly improved the manuscript, making it a more solid contribution to the field of biocatalysis, particularly within the underdeveloped area of α -oxoamine synthases (AOS).

The authors have provided additional HPLC and MS data to support their findings, strengthening the evidence for the formation of targeted pyrrole products. Although the isolation of individual amino ketones remains challenging, their acknowledgment of these limitations is clear, and the additional data provided offers a reasonable alternative.

Their explanation regarding the stereochemical analysis is well-justified, particularly in the context of producing planar pyrrole products, where stereochemistry is not relevant. The decision to remove chiral labels from figures is appropriate given this focus.

While the authors did not employ Differential Scanning Fluorimetry (DSF), their conventional thermostability tests using enzyme activity retention after incubation at elevated temperatures are sufficient for comparing the new ThAOS variants to the wild-type enzyme. The method they used is well-established and provides practical insight into the optimal conditions for biocatalysis.

The manuscript has been restructured for better flow, reducing redundancy and improving clarity. The authors have also addressed the grammatical and language issues noted, leading to a more polished and accessible presentation.

The authors have taken all minor suggestions into consideration, including clarifying acronyms, improving figure legends, and correcting formatting inconsistencies. These adjustments enhance the readability and professionalism of the manuscript.

Based on the comprehensive revisions and the authors' careful attention to feedback, I believe this manuscript is now well-prepared for publication. The additional data, clarified methodology, improved organization, and enhanced language quality have strengthened the manuscript's contribution to the field.

Additionally, I provide my assessment of the authors responses to other comments raised by reviewers.

1. Do the ketone products racemize under the reaction conditions? :

The authors have provided a reasonable response, indicating that such a detailed enzymatic study is beyond the scope of this manuscript. Their explanation that the stereochemistry becomes irrelevant due to the planar pyrrole products formed is logical. I find this response satisfactory, as it aligns with the focus of the study.

2. Evaluation of the substrate scope with HPLC, MS, and NMR data.

This is similar to my request and in response the authors have included HPLC and MS data for five pyrrole derivatives and characterized pyrrole 1 using NMR, addressing this concern effectively. While isolating and analyzing all intermediates would strengthen the study, the provided data is sufficient for the scope of this work.

3. Detailed table of contents in the Supporting Information.

The addition of a detailed table of contents is a welcome improvement, as it enhances the clarity and navigability of the Supporting Information.

4. Capitalization and formatting of L/D amino acid.

The authors have corrected all instances of amino acid notation throughout the manuscript.

5. References for methods.

The authors have clarified their use of appropriate references for the methods, such as the DTNB assay, and cited these accordingly.

6. Formatting issues in the Methods section and NMR assignments and description of methods.

The authors have addressed the formatting issues in the Methods section. The revised Methods section now includes more detailed descriptions of the reagents and materials used.

Reviewer #4

(Remarks to the Author)

I co-reviewed this manuscript with one of the reviewers who provided the listed reports. This is part of the Communications Chemistry initiative to facilitate training in peer review and to provide appropriate recognition for Early Career Researchers

who co-review manuscripts.

6th November 2024

Ashley et al, COMMSCHEM-24-0221

Dr. Huijuan Guo, Senior Editor, Communications Chemistry

Dear Huijuan,

Many thanks for your patience with respect to this manuscript.

We thank the referees for the valuable and insightful comments. We have tried our best to answer each comment and included additional data where required. All changes to the original manuscript and SI are highlighted in yellow.

Overall, key points include:

We have synthesised the key product pyrrole (1) and include NMR data on this. We have also moved some data from the SI (e.g. Table 2) to the main text to highlight the improved substrate scope of the engineered *ThAOS V79S* biocatalyst. One referee has asked us to characterise each aminoketone product and asked if the stereochemistry of the initial amino acid is retained. We felt that this was not required since we carry out a KPR to generate the planar product pyrrole. We also cite key papers on the catalytic mechanism of the AOS family (e.g. ALAS, SPT, AONS) that discuss the retention of the stereochemistry of the amino acid substrate. We do include the derivitization of the glycine aminoketone product. As also requested, we have re-drawn key data such as Fig. 5 to make it much clearer to show product formation and the internal standard.

Where ever possible we have taken advice on contracting the text, removing non essential comments e.g. we shorten the structural discussion and move text to the SI . We have also given a break down of the background studies of members of the AOS family e.g. references 15-32 and commented on each enzyme/paper.

We now submit a significantly re-worked manuscript and hope is satisfactory for publication.

Reply to referees' comments

Referee #3

This referee provided their comments as a separate PDF file. We have addressed their comments/suggestions below.

This referee thinks the “manuscript presents an interesting contribution to the underdeveloped family of oxoamine synthases” and the “novelty lies mainly in the introduction of mutations that offer expanded scope”.

They make a number of useful points and comments that we have addressed in the re-submitted manuscript.

They state that the substrate scope analysis offers indirect evidence for the formation of the targeted products. Only in one case, they report the NMR of the cascade reaction. The authors should provide more NMR or GC-MS to prove the formation of the α -amino ketones.

Ideally, we would have isolated the amino ketones, however, they are notoriously difficult to isolate at scale. We have found only 5 commercially available, relatively inexpensive α -amino ketones. One

could imagine being formed by 4 members of the AOS enzyme family; glycine aminoketone by KBL, AON from AONS, ALA from ALAS, KDS from SPT. The aminoacetophenone (AAP) is also available.

We have previously monitored the formation of AON (the aminoketone from L-Ala and pimeloyl-CoA) in our recent paper on the fusion enzyme BioWF (Richardson et al., ChemBiochem 2022). However our goal was the production of pyrroles through the 2nd KPR reaction and we provided data to support formation of this target.

Table I. The referee would like us to determine the e.e. of the AON product of the ThAOS reaction. Again, like in our answer above, we would argue that this is outwith the scope of our intended formation of a planar/flat pyrrole where the stereochemistry of the original amino acid substrate is lost. In the AOS consensus mechanism (Fig. 2) we cover the known stereochemistry of the AOS family; they have been shown to be stereo-retentive i.e. the S-stereochemistry of the amino acid is retained in the product aminoketone – a double inversion, decarboxylative, Claisen condensation - this has been shown in studies with ALAS, SPT and AONS (cite these). We have now explained this in more detail and feel that a stereochemical analysis is redundant here. To that end since we have determined the stereochemistry of all the aminoketone products we have removed the chirality from the ChemDraw figures.

This referee suggest that can measure the thermostability of the ThAOS varaints using a qPCR system /DSF. We are sorry but we don't have access to such an instrument and have relied on the activity of the enzyme remaining after incubation at various elevated temperature for 0-120 mins. This method is conventionally used and is more practical in terms of giving the optimum temperature at which the biocatalysts can be used. This allowed us to compare the new ThAPOS variants with the WT that we reported in Ashley et al., 2023.

The referee asks us to re-order the structure and shorten the manuscript. We have done this and highlight the changes in the new manuscript.

They also ask us to improve the English and correct the grammatical and spelling mistakes. Thanks for listing some of these, we have proofed the new manuscript and spent time removing any errors.

We appreciate this referee's other minor points – we have taken all of these into consideration and, where appropriate, have re-worked these into the manuscript.

Reviewer #1 (Remarks to the Author):

In this manuscript, the authors demonstrated the structure-guided engineering of ThAOS to achieve variants with broader scope of substrates. The crystal structure of the ThAOS V79A mutant with a bound PLP:penicillamine external aldimine ligand, was investigated. I find it interesting.

I recommend to revise some points (see below).

1. Do the ketone products racemize under the reaction conditions?

This is an interesting question but it is one that has not been carried out with the best characterised enzymes in the AOS family e.g. ALAS with product ALA, SPT with product 3-KDS and AONS with product AON. We felt that such a detailed enzymatic study is outwith the scope of this paper and as we answered referee 3 above, we generated planar pyrrole products by coupling the AON products formed by the ThAOS variants with the KPR.

2. In the evaluation of the substrate scope, the HPLC and MS analysis of the products with various substrates should be provided. At least some of the products should be purified and provide the NMR data.

This is a similar question to referee 3. Rather than the aminoketone, we now provide HPLC and MS data on five of the pyrroles derived from L-Cpg, D/L-Alg, L-Pra, L-Ala and Gly (pyrrole 1, which we have characterised by NMR).

3. It would be more clear if you provide a detailed table of content at the beginning of the supporting information.

Thanks, we have done this.

4. All instances of L or D in front of amino acid should be capitalized and adjust to the smaller size. For example, in line 272. Please check the whole manuscript.

Yes, we have done this.

5. Please mention the reference for each method used in this study.

We are not sure what the referee means here but have cited each method as appropriate e.g. the DTNB assay for measuring CoASH formation that we published Raman et al., 2009.

6. Please check the text format in the Method section. The sub-title should be capitalized for each word. The superscript and subscript format in NMR section should be corrected. The NMR data assignment to the specific position should be marked in the structure.

We have changed this as appropriate.

7. The description of materials in the method section is obscure. Please reveal more detailed information.

We have changed these and added a lot more detail about the source and types of reagents.

Reviewer #2 (Remarks to the Author):

The manuscript 'Rational engineering of a thermostable a-oxoamine synthase biocatalyst expands the substrate scope and synthetic applicability' describes the application of PLP-dependent enzyme ThAOS in the biocatalytic decarboxylative Claisen-like condensation reactions between amino acid and thioester substrates. This article builds off of the Campopiano lab's previous studies adapting a CoA cofactor recycling system and chemoenzymatic production of a-aminoketones and pyrroles. A variety of interdisciplinary techniques are used to better understand this C-C bond forming enzymology, including: UV-spectroscopy, substrate engineering, site-saturation mutagenesis, X-ray crystallography, biocatalysis and chemoenzymatic synthesis. I appreciated the effort put into this manuscript and found it to be a very dimensional study into a chemically useful enzyme family. One area of particular interest was the spectroscopic variation of the different mutants on improving the quinonoid species in the presence of the second thioester substrate. I have a few suggestions to improve this manuscript – however, following these adjustments, I believe this manuscript is appropriate for publication in Communications Chemistry.

Major comments:

DTNB assay vs. product output – The authors have nicely shown the use of the DTNB reagent to quantify free sulfhydryl groups and show amino acid and CoA/SNAC substrates that work with ThAOS and mutants. However this does not directly measure a-aminoketone product formation and could result from non-specific hydrolysis of the thioester. Have the authors quantified the DTNB output compared to product formation? Alternatively, has this analysis been done before for this enzyme family in literature to show that the DTNB read out is an appropriate substitute for direct product quantification? Some insight here would help reinforce the utility of this method and give additional confidence to the results for all ThAOS variants presented throughout.

Thanks for this suggestion. This comment is in line with the two others. We introduced the convenient DTNB assay to measure CoASH formation for our study of the AOS family enzyme SPT in Raman et al, 2009. This method has been used by many other groups and been cited >70 times. There we showed that CoASH formation correlates with aminoketone product AON formation. We also recently studied a BioWF fusion enzyme (BioF is the AOS domain) and used both the DTNB assay and LC-MS analysis to measure product formation (Richardson et al., ChemBioChem 2022).

Figure 5 alignment – It took me a while to realize that the different temperature traces were staggered at an angle – I would encourage the authors to line things up on the same x-axis or use shading to make it obvious which peaks correspond to the acetyl-SNAC, I.S., and 1 in each trace.

We are sorry this was not clear, we have re-drawn this data and shaded/annotated as appropriate. We hope this is clearer.

Minor comments:

Page 1, line 38 – It felt a bit heavy handed associating refs 15-32 with one sentence in the introduction – potentially breaking them up would be more helpful to delineate their relative contributions.

Sorry, about this – we have broken these up and grouped them.

Figure 1 – Including a citation to sections B and C would further connect them to their literature references. I enjoyed the use of the red highlight for the new C-C bond in Panel A – continuing that in panels B through D would further emphasize the impact of this enzymology.

Thanks for this suggestion, we have done this.

Page 3, lines 93-96 – Please define KBL, AONS, DTNB, SPT, and other acronyms on their first introduction to help improve transparency to non-expert audiences.

Sorry about this, we have done this.

Figure 2 – There is another mechanistic step between the b-ketoacid intermediate and the product aldimine – either depict the enolate intermediate (and a-carbon reprotonation) or show more than one arrow between these intermediates. Quinonoid is also misspelled in the figure legend.

We have added an extra arrow to convey that there are two steps.

Figure 3, line 185 – the PDBID for ThAOS is missing in the figure caption.

Thanks, we have added this.

Page 9, Table 1 – Please include a legend for the grey scale shading in the heat map to give insight into the turnover numbers for each variant/amino acid pair.

Thanks for this suggestion, we have changed the colours and made them blue and added a scale based on enzyme activity.

Page 9, lines 368 and beyond – Only include integer numbers for yields

Thanks, we have done this.

Page 10, line 388 – Presumably missing reference here in place of (Ashley)

Thanks, have fixed this.

Page 13, lines 490 – 503 – The alpha center stereochemistry for L-penicillamine is not depicted in panel C, and the Figure caption for this panel is missing.

Thanks, we have fixed this.

References – Refs 35 and 36 are duplicated

Sorry, we noticed this after submission and fixed it. We also noted refs 28 and 43 are duplicates, now fixed. We have worked to upgrade and improve the whole bibliography with relevant citations.